# Identification of Viruses Infecting *Phalaenopsis* Orchids Using Nanopore Sequencing and Development of an RT-RPA-CRISPR/Cas12a for Rapid Visual Detection of Nerine Latent Virus

**DOI:** 10.3390/ijms25052666

**Published:** 2024-02-25

**Authors:** Hyo-Jeong Lee, Hae-Jun Kim, In-Sook Cho, Rae-Dong Jeong

**Affiliations:** 1Department of Applied Biology, Institute of Environmentally Friendly Agriculture, Chonnam National University, Gwangju 61185, Republic of Korea; hjhjhj8@naver.com (H.-J.L.); wns980508@naver.com (H.-J.K.); 2Horticultural and Herbal Crop Environment Division, National Institute of Horticultural and Herbal Science, Rural Development Administration, Wanju 55365, Republic of Korea; tuat@korea.kr

**Keywords:** *Phalaenopsis* orchids, nanopore sequencing, RT-RPA-CRISPR/Cas12a, visualized diagnosis, rapid detection

## Abstract

*Phalaenopsis* orchids are one of the most popular ornamental plants. More than thirty orchid viruses have been reported, and virus-infected *Phalaenopsis* orchids significantly lose their commercial value. Therefore, the development of improved viral disease detection methods could be useful for quality control in orchid cultivation. In this study, we first utilized the MinION, a portable sequencing device based on Oxford Nanopore Technologies (ONT) to rapidly detect plant viruses in *Phalaenopsis* orchids. Nanopore sequencing revealed the presence of three plant viruses in *Phalaenopsis* orchids: odontoglossum ringspot virus, cymbidium mosaic virus, and nerine latent virus (NeLV). Furthermore, for the first time, we detected NeLV infection in *Phalaenopsis* orchids using nanopore sequencing and developed the reverse transcription–recombinase polymerase amplification (RT-RPA)-CRISPR/Cas12a method for rapid, instrument-flexible, and accurate diagnosis. The developed RT-RPA-CRISPR/Cas12a technique can confirm NeLV infection in less than 20 min and exhibits no cross-reactivity with other viruses. To determine the sensitivity of RT-RPA-CRISPR/Cas12a for NeLV, we compared it with RT-PCR using serially diluted transcripts and found a detection limit of 10 zg/μL, which is approximately 1000-fold more sensitive. Taken together, the ONT platform offers an efficient strategy for monitoring plant viral pathogens, and the RT-RPA-CRISPR/Cas12a method has great potential as a useful tool for the rapid and sensitive diagnosis of NeLV.

## 1. Introduction

*Phalaenopsis* orchids, commonly known as “moth orchids”, are a popular and visually stunning group of orchids known for their exquisite and long-lasting blooms. They are native to Southeast Asia, including countries such as Indonesia, Malaysia, the Philippines, and parts of Australia, and have become one of the most widely cultivated and traded orchid varieties worldwide [1]. The combination of cultural significance, changing lifestyles, and effective marketing strategies has led to an increase in the demand for *Phalaenopsis* orchids in Korea. As the appreciation for these graceful and long-lasting flowers continues to grow, the market for *Phalaenopsis* orchids is likely to remain strong in the country [2]. *Phalaenopsis* orchids are typically propagated using tissue culture techniques and are commonly cultivated in densely planted greenhouses in Korea. However, this cultivation approach can easily spread viral diseases [3].

Viruses have the potential to infect entire orchid plants under cultivation and may even spread to affect the majority of orchids within the entire cultivation area. The commercial value of orchids can significantly decline when they are infected by viruses. Therefore, the development of improved viral disease detection methods is a critical task in ensuring the quality control of orchid cultivation [3]. The most common viruses infecting *Phalaenopsis* orchids worldwide are odontoglossum ringspot virus (ORSV) and cymbidium mosaic virus (CymMV) [4]. ORSV is known to cause streaks, stripes, mosaic patterns, or ringspots on orchid leaves, while CymMV induces chlorotic or necrotic symptoms on leaves and flowers. When these viruses co-infect orchids, the resulting symptoms are often more severe, as demonstrated in previous studies [5]. Such viral infections are frequently observed on *Phalaenopsis* orchid farms in Korea. In fact, a high mixed infection rate was reported, with 83.6% of in vitro plantlets from 45 cultivars and 92.5% of potted plants from 21 cultivars in Korea in 2021, greatly reducing their commercial value [6]. Rapid and accurate virus detection enables the swift identification and removal of diseased *Phalaenopsis* orchids. Early removal is crucial in preventing a decline in marketability caused by virus infections and ensuring the maintenance of plant quality. The most routinely used detection methods for orchid viruses involve nucleic acid-based amplification assays such as reverse transcription–polymerase chain reactions (RT-PCRs), TaqMan probe-based quantitative real-time PCR, and reverse transcription–loop-mediated isothermal amplification (RT-LAMP) [6,7,8]. 

High-throughput sequencing (HTS), also known as next-generation sequencing, has revolutionized the field of plant virology by providing a powerful and efficient tool for the detection, characterization, and genomic study of plant viruses [9]. Nanopore sequencing is a cutting-edge DNA sequencing technology that allows for the real-time analysis of individual DNA molecules as they pass through a tiny pore or nanopore. It is known for its portability, speed, and the ability to produce long reads. Nanopore sequencing is primarily utilized for the analysis of DNA, RNA, and even proteins [10]. Its versatility makes it suitable for various applications in virology, epidemiology, and diagnostics, contributing to our ability to detect and respond to viral infections more effectively [11]. Its ability to provide long-read sequencing data is particularly advantageous for studying the complex genomes of plant viruses and understanding their impact on agricultural ecosystems [12,13].

CRISPR, which stands for Clustered Regularly Interspaced Short Palindromic Repeats, is a revolutionary technology primarily recognized for its gene-editing capabilities. However, it has also been adapted for detection purposes, giving rise to a technique known as CRISPR-based detection or CRISPR-based diagnostics. This approach leverages the precision and programmability of the CRISPR system to identify specific DNA or RNA sequences, making it a powerful tool for detecting pathogens, genetic mutations, and various other biomolecules [14]. CRISPR/Cas12a-based virus detection is a highly sensitive and specific molecular diagnostic technique that utilizes the CRISPR/Cas12 system to identify the presence of a target virus. This approach relies on the collateral cleavage activity of Cas12, an enzyme associated with the CRISPR system. The binding of a specific guide RNA to a complementary target sequence triggers Cas12, leading to the cleavage of nearby single-stranded DNA molecules [15,16]. This technique offers several advantages, including high specificity, sensitivity, and the potential for rapid results. It has already been used in the development of diagnostic tests for plant viruses. Consequently, this technique holds promise for accurate and timely virus detection in both field and research settings [17]. 

In this study, electron microscopy and nanopore sequencing were utilized to identify the virus in symptomatic *Phalaenopsis* orchids. Nanopore sequencing emerges as a promising strategy for efficiently monitoring plant viral pathogens, offering a foundation for accurately diagnosing viral co-infections. We used nanopore sequencing technology to initially identify nerine latent virus (NeLV) in *Phalaenopsis* orchids worldwide. Furthermore, we developed a highly specific and sensitive NeLV detection assay based on CRISPR/Cas12a, which incorporates recombinase polymerase amplification (RPA) for use in *Phalaenopsis* orchids. Subsequently, we validated the performance of this assay using field-collected samples of *Phalaenopsis* orchids. The developed CRISPR/Cas12a-based assay offers a convenient method for conducting comprehensive surveys of NeLV in the field.

## 2. Results

### 2.1. Observation of Viral Particles Using Transmission Electron Microscopy (TEM)

Plant virus particles were analyzed in *Phalaenopsis* orchid samples using transmission electron microscopy (TEM). We identified virus particles with similar shapes to those reported for CymMV, which is a filamentous virus with a length of 480–500 nm; ORSV, which is a rod-shaped virus with a length of 290–300 nm; and NeLV, which is a filamentous virus with an average length of 660 nm (Figure 1B).

### 2.2. Identification of Viruses by Nanopore Sequencing

MinKNOW was used to analyze leaf samples showing symptoms of leaf mosaic patterns, yellowing, and necrosis (Figure 1A). The analysis produced a total of 528,819 reads with an average read length of 614.2, utilizing the Flongle flow cell. Of these, 97,476 were obtained after adapter trimming and quality filtering (>Q7), with an average trimmed read length of 87.8 (Table 1).

The WIMP workflow was used to rapidly identify and quantify species in metagenomic samples. To identify viruses in each library, the trimmed reads obtained from ONT sequencing were subjected to BLAST analysis in NCBI VirDB. This analysis revealed the presence of three viruses, namely odontoglossum ringspot virus (ORSV), cymbidium mosaic virus (CymMV), and nerine latent virus (NeLV), within the moth orchid.

### 2.3. Genome Assembly

Genome assembly was carried out using the reference genomes of three species, namely ORSV, CymMV, and NeLV, which were isolated from the moth orchid. This process resulted in nearly complete sequences with coverage ranging from 94.1% to 100%. ORSV, obtained from the moth orchid, exhibited 2573 reads mapped with a 92.3% identity. CymMV showed 12,913 reads mapped at a 92.6% identity, and NeLV displayed 1466 reads mapped at a 97% identity. The consensus sequences of the CP from each virus isolated from the moth orchid were deposited in GenBank (ORSV, LC790454; CymMV, LC790455; and NeLV, LC790456) (Figure 2 and Table 2).

### 2.4. Phylogenetic Analyses

Phylogenetic tree analyses were performed on three viruses identified from *Phalaenopsis* orchids. The phylogenetic tree based on the CP sequence of CymMV was divided into two separate trees, revealing its closest relation to the South Korea isolate (AB541550) from *Dendrobium*. Similarly, NeLV was also divided into two phylogenetic trees, showing its closest relation to the South Korea isolate (MZ927426) from *Narcissus*. Lastly, ORSV was divided into two phylogenetic trees, indicating its closest relation to the China isolate (KF836066) from *Phalaenopsis* (Figure 3).

### 2.5. Validation of the Identified Viruses

To validate the presence of the identified viruses, RT-PCR was performed using primers known to be specific to each virus. The RT-PCR results were consistent with the viruses identified through ONT sequencing, and the amplicon sizes matched those expected from each specific primer (Figure 4). Subsequently, Sanger sequencing was conducted on the RT-PCR amplification products, confirming the presence of each virus through a BLASTn analysis.

### 2.6. Optimization of Reaction Temperature and Time

To determine the optimal temperature and time for NeLV detection using RT-RPA-CRISPR/Cas12a, we conducted experiments at 34, 36, 48, 40, 42, 44, and 48 °C for various durations. Both the optimal temperature and time were determined by visually observing the fluorescence emitted during the Cas12a cis cleavage of the RT-RPA product, guided by sgRNA, and the trans cleavage of the ssDNA probe through the CRISPR/Cas12a cleavage system. Amplification was observed at all temperatures, with the highest fluorescence intensity observed at 34 °C (Figure 5A). Then, to determine the optimum reaction time, we performed RT-RPA-CRISPR/Cas12a reactions for 1, 3, 5, 10, 15, 20, 25, and 30 min at the optimal temperature of 34 °C. Amplification was observed at 1 min, while high fluorescence intensity was confirmed at 3 min (Figure 5B). These results were confirmed using UV lights and blue lights. Therefore, we selected 34 °C and an incubation time of 3 min for further RT-RPA-CRISPR/Cas12a experiments.

### 2.7. Specificity and Sensitivity

To assess the specificity of the RT-RPA-CRISPR/Cas12a method, we used samples infected with CymMV, ORSV, TSWV, and CMC, which are commonly found in *Phalaenopsis* orchids. The results showed that only NeLV produced a significant fluorescent signal, whereas the remaining viruses and negative controls (healthy samples and NTC) yielded negative results without any cross-reactivity. These observations indicate that the RT-RPA-CRISPR/Cas12a technique exhibits excellent specificity, as it does not detect other viruses (Figure 6A).

To ensure the specificity of the crRNAs of the CRISPR/Cas12a system, we evaluated the specificity using three different crRNAs that did not target any sequence in the NeLV genome. As a result, a fluorescent signal was observed only from the specific crRNA of NeLV, and these results demonstrated that any signal or detection was due to specific targeting of the functional crRNA and not the non-specific activation of the Cas enzyme (Appendix A).

To evaluate the sensitivity of NeLV detection, we performed the optimized RT-RPA-CRISPR/Cas12a method to detect serially diluted transcripts ranging from 100 fg/μL to 10 zg/μL. The fluorescence intensity emitted by RT-RPA-CRISPR/Cas12a gradually decreased as the concentration of diluted transcripts decreased, and we were able to detect fluorescence intensity at a concentration of 10 zg/μL (Figure 6B). In comparison, conventional RT-PCR did not show any amplification at concentrations below 1 ag/μL. These results indicate that the sensitivity of the RT-RPA-CRISPR/Cas12a assay developed for NeLV detection is 1000-fold higher than that of conventional PCR.

### 2.8. Validation of CRISPR/Cas12 Using Field Samples

To verify the feasibility of employing RT-RPA-CRISPR/Cas12a for NELV detection in field samples, we compared it to RT-PCR using 21 *Phalaenopsis* orchid samples and 17 daffodil (*Narcissus tazetta*) samples. The results of the field test comparison showed that among the 21 *Phalaenopsis* orchid samples, RT-PCR detected three cases, whereas RT-RPA-CRISPR/Cas12a detected 10 cases. In the case of the 17 daffodil samples, RT-PCR identified 10 cases, while RT-RPA-CRISPR/Cas12a detected 12 cases (Figure 7 and Table 3). Therefore, the results of the field sample testing demonstrate that RT-RPA-CRISPR/Cas12a is more suitable and sensitive for practical field applications compared to RT-PCR.

## 3. Discussion

*Phalaenopsis* orchids are renowned for their enduring and captivating flowers, making them a popular choice among ornamental plants [18]. However, viral infections in these orchids can lead to significant economic losses by diminishing the quality of the flowers. Therefore, the availability of reliable and precise detection methods for early viral pathogen diagnosis is crucial. Immunoassays or PCR-based methods have limitations in detecting viral pathogens, primarily because they can only identify specific known targets and may overlook new or uncommon viral strains or variants. In addition, these methods require prior knowledge of the target virus and struggle to detect multiple viruses simultaneously in a single test [19].

HTS is a very powerful technique for accurately identifying viruses in plant samples without previous knowledge of viral sequences. Although HTS technologies can generate large data sets, sample preparation and actual sequencing are time-consuming and sequencing equipment is often expensive and bulky, limiting their utility in agricultural fields. On the other hand, the ONT platform has proven successful in identifying multiple plant viruses in a single sample at relatively low cost and in a short period of time. In a single run, the system can perform the real-time detection and diagnosis of multiple virus species, showcasing significant potential for various applications [20]. The ONT platform has emerged as a dependable, real-time, and portable technology, significantly advancing the study and monitoring of diverse viral pathogens, including human and other animal viruses. Similar to other HTS methods, benchtop sequencers find routine use in laboratories because ONT-based studies, like those using the MinION, require the use of advanced bioinformatics tools for data management and analysis [21]. However, the ONT platform has had limited applications in the field of plant virology thus far [22]. Therefore, we assessed the ONT platform’s potential for virus detection and characterization, demonstrating that nanopore sequencing holds promise as an alternative method for identifying viral pathogens in plants. The ONT platform utilizes massive parallel sequencing technology, offering the potential for comprehensive genome studies of both known and unknown viruses. Despite advancements, the ONT platform still encounters challenges in achieving high single-base accuracy [23,24]. Nevertheless, recent progress in nanopore equipment, reagents, and analytical algorithms, notably in base calling, has resulted in a consensus accuracy exceeding 99.9% [25,26]. While Flongle flow cells offer significantly lower throughput compared to standard flow cells, their cost-effectiveness renders them attractive, particularly in scenarios where extensive datasets for plant virus diagnostics are unnecessary [26].

Three viruses were identified in *Phalaenopsis* orchids through the utilization of the ONT platform. Among them, ORSV and CymMV are recognized as the most detrimental viruses affecting *Phalaenopsis* orchids [27]. Previous investigations have delineated the disease manifestations associated with these viruses—chlorotic and necrotic spots on leaves infected with CymMV—while leaves co-infected with ORSV exhibit symptoms such as yellow stripes or mosaic patterns [3,5]. Notably, these symptoms are consistent with those observed in samples analyzed using the ONT platform. NeLV, isolated from *Phalaenopsis* orchids, is documented to infect various *Amaryllidaceae* species and was initially encountered during the quarantine of *Hippeastrum* bulbs in Korea [28,29,30]. While NeLV has not been previously documented in *Phalaenopsis* orchids worldwide, this study marks its inaugural report. 

In this study, we developed a diagnostic technique that combines the RT-RPA and CRISPR/Cas12a platforms to establish a robust and efficient method for detecting NeLV, initially identified in *Phalaenopsis* orchids. Isothermal amplification plays a pivotal role in CRISPR-based detection schemes [31]. Hence, we implemented an RT-RPA method for NeLV, opting for the RPA technique due to its minimal primer requirement and rapid amplification kinetics. The RT-RPA-CRISPR/Cas12a method offers numerous advantages over other molecular diagnostic techniques employed for NeLV detection. RT-RPA-CRISPR/Cas12a reactions can be amplified under low isothermal conditions, requiring only equipment capable of fluorescence measurement, thereby obviating the need for expensive specialized instrumentation such as thermocyclers [32]. These benefits can be leveraged by utilizing body heat or an affordable hand warmer [32,33,34]. Furthermore, the entire diagnostic procedure is completed in just 1 min for the RT-RPA reaction and 15 min for the CRISPR/Cas12a system, following RNA extraction (Figure 8). This rapid detection capability surpasses traditional RT-PCR methods. Occasionally, the general RPA reaction may yield non-specific amplification due to non-specific binding, resulting in false positive signals. However, this issue is effectively mitigated by initially recognizing and amplifying the target using RPA, followed by the recognition and cleavage of the amplified RPA product with CRISPR/Cas12a [33].

Due to its facile target-dependent programmability, CRISPR/Cas has emerged as a tool for combating viruses across diverse organisms. With the proliferation of Cas proteins exhibiting varied activities, the CRISPR/Cas system has found utility in the diagnosis of infectious agents, including viruses [35,36,37,38,39]. These Cas proteins exhibit specificity towards a spectrum of nucleic acids [17]. Notably, the Cas13 protein necessitates a transcription step for RNA generation as it targets ssRNA, albeit false positives may arise due to the utilization of a reporter for ssRNA [40]. Similarly, the Cas14 protein, tailored to target single-stranded DNA (ssDNA), mandates the generation of ssDNA from the amplification product of double-stranded DNA (dsDNA) through the utilization of modified primers and T7 exonuclease [41]. To overcome these challenges, the present study opted for Cas12a, characterized by its recognition of the protospacer adjacent motif (PAM) sequence and its ability to cleave both dsDNA and ssDNA.

The RT-RPA-CRISPR/Cas12a system operates by cleaving the sequence-specific RPA product of NeLV using the guide RNA during the reaction. Subsequently, Cas12a induces random cleavage of the single-stranded DNA reporter, resulting in an amplified fluorescence signal. Employing virus-specific RPA primers enables the detection of NeLV without encountering cross-reactivity in samples potentially infected with other viruses. This indicates that the RT-RPA-CRISPR/Cas12a detection assay is highly specific and can be effectively used for RNA-based virus detection. Furthermore, the RT-RPA-CRISPR/Cas12a assay exhibited remarkable sensitivity, as it successfully detected NeLV transcripts even at zeptogram concentrations. The feasibility of utilizing RT-RPA-CRISPR/Cas12a was evaluated using *Phalaenopsis* orchid and daffodil samples harboring NeLV, wherein the results demonstrated high accuracy, indicative of its potential for early NeLV diagnosis.

## 4. Materials and Methods

### 4.1. Sample Collection and RNA Extraction

*Phalaenopsis* orchids showing symptoms of chlorotic flecks, mosaics, and necrosis were collected from Samcheok, Republic of Korea, in August 2021 and stored at −80 °C until use. Total RNA was extracted using the BeniPrep^®^ Super Plant RNA extraction kit (InVirusTech Co., Gwangju, Republic of Korea) according to the manufacturer’s instructions. The concentration and quality of the extracted RNA were measured using a BioDrop spectrophotometer (Biochrom, Ltd., Cambridge, UK). For nanopore library preparation, pure and enriched RNAs were obtained using the method described in a previous study [13]. The ribo-depleted RNA was then quantified using a Qubit™ 4 Fluorometer with a Qubit™ RNA HS Assay Kit and stored at −80 °C until use.

### 4.2. Observation of Viral Particles Using Transmission Electron Microscopy (TEM)

Virus particles were observed using TEM in *Phalaenopsis* orchid samples using the direct negative staining method. *Phalaenopsis* orchid leaf samples were mixed with phosphate buffer in a 1:1 ratio and incubated at room temperature for 10 min. The mixture was centrifuged at 12,000 rpm for 20 min and the supernatant was used for TEM analysis. The prepared samples were fixed on copper grids (Electron Microscopy Sciences, Hatfield, PA, USA), negatively stained with 1% ammonium molybdate (Sigma-Aldrich, St. Louis, MO, USA), and analyzed for virus particles using TEM (200 kV; JEOL Ltd., Tokyo, Japan) at the Center for Research Facilities of Chonnam National University.

### 4.3. Nanopore Library Preparation and ONT Sequencing

The library was generated using a direct cDNA sequencing kit (SQK-DCS109) in combination with a Flongle Starter pack (Oxford Nanopore Technologies, Oxford, UK) following the manufacturer’s guidelines and previously described methods [13,26]. The library mixture was loaded into an R9.4.1 Flongle flow cell (FLO-FLG001), and sequencing was performed on a MinlON device by Oxford Nanopore Technologies. The flow cell underwent a 24 h run, managed through the MinKNOW software (version 21.11.7). Real-time base calling of the resulting reads was conducted using the high-accuracy mode (quality score of 7) in Guppy (version 5.1.12) within the MinKNOW software.

### 4.4. Bioinformatic Analysis

The base-called FASTQ files were converted to FASTA files using Geneious Prime (version 21.1.1), and adapter sequences were subsequently removed from the trim end. To validate the sequences of plant viruses, a combination of methods was employed. This included using the FASTQ “What’s in my Pot?” (WIMP) workflow in EPI2ME Agent (version 3.4.2) and conducting a BLASTn search (-outfmt 6) against the National Center for Biotechnology Information (NCBI) Viral Genome data, specifically the Viral RefSeq Database (VirDB) (version 21.11.4). For the identification of plant virus sequences, the sequencing reads were aligned with plant virus genomes exhibiting significant BLAST scores, as determined by top hits obtained from NCBI BLASTn searches. This alignment was achieved using the “Map to Reference” tool in Geneious Prime with default settings, specifically Medium Sensitivity/Fast, ultimately resulting in consensus sequences. 

### 4.5. Phylogenetic Analysis

For phylogenetic analysis, the complete coat protein (CP) gene sequences of identified viruses were aligned with CP gene sequences of other isolates obtained from NCBI BLASTn. The alignment was performed using the BioEdit Sequence Alignment Editor (version 7.0.5.3) with ClustaIW Multiple alignment. Subsequently, maximum likelihood phylogenetic trees were generated using MEGAX (version 10.2.4), employing the Tamura–Nei model of nucleotide substitution as described [42,43]. The robustness of the phylogenetic tree was assessed by calculating bootstrap values based on 1000 replicates. 

### 4.6. RT-PCR for Validation

RT-PCR was employed to confirm the presence of plant viruses identified through ONT sequencing in each library. The primers specified in Appendix A were used for this assay. Total RNA was amplified using the SuPrimeScript RT-PCR premix (Genet Bio, Daejeon, Republic of Korea), following the manufacturer’s protocols. All RT-PCR products were sequenced using Sanger sequencing, and the sequences were subsequently verified using a BLASTn search.

### 4.7. Data Availability

The FASTQ files produced for each library in this study have been archived in the NCBI Sequence Read Archive (SRA) repository within the NCBI BioProject database, while the plant virus sequences that were identified have been deposited in GenBank, each assigned a unique accession number. 

### 4.8. RT-RPA Primers’ Design and crRNA Preparation

The RT-RPA primers were meticulously designed within the highly specific CP region of the NELV using PrimedRPA [44] and following the instructions provided in the TwistDX RPA manual (TwistDX, Ltd., Cambridge, UK). 

The CRISPR RNA (crRNA) targeting the region adjacent to the protospacer adjacent motif (PAM) site in the RT-RPA amplified fragment of NeLV was designed following established design principles. To synthesize the crRNA, T7-crRNA-F and NeLV-crRNA were initially denatured for 5 min and subsequently annealed by gradually reducing the temperature in 1 °C decrements from 95 °C to 20 °C. The crRNA was then synthesized using the annealed oligo as a template, following the instructions provided by the TranscriptAid T7 High Yield Transcription Kit (Thermo Fisher Scientific, Waltham, MA, USA), with an incubation at 37 °C for 4 h. The synthesized crRNA was purified using an RNA Clean and Concentrator-5 kit (Zymo Research, Irvine, CA, USA), and the concentration of the purified crRNA was quantified using a Qubit^TM^ 4 Fluorometer with a Qubit^TM^ RNA HS Assay Kit (Thermo Fisher Scientific). The crRNA was subsequently used at a final concentration of 1 μM. 

### 4.9. Preparation of Standard Virus Transcripts

Primer sets designed for the NeLV-CP gene were used to generate cDNA for subsequent in vitro transcription. The resulting amplicon was then cloned using the T&A cloning kit (Yeastern Biotech, Taiwan) following the provided protocol. In vitro transcription was conducted using T7 RNA polymerase (Promega, Madison, WI, USA). The concentration of the NeLV-CP transcript was determined using a spectrophotometer (BioDrop), and the transcript was stored at –80 °C for further experimentation. 

### 4.10. RT-RPA-CRISPR/Cas12a Assays

The RT-RPA reactions were carried out using the TwistAmp Basic Kit (TwistDx, Cambridge, UK). Briefly, a reaction mixture was prepared with a final volume of 50 μL, consisting of 4.8 μL each of forward and reverse primers (10 μM), 1.2 μL of recombinant RNase inhibitor (40 U/μL) from Takara (Kyoto, Japan), 1 μL of M-MLV reverse transcriptase from Promega (Madison, WI, USA), lyophilized enzyme powder, DEPC-treated water, and RNA. Subsequently, MgOAc (280 mM) was added, followed by incubation at 42 °C for 30 min.

The CRISPR/Cas12a reactions were conducted using the following components: 2.5 μL of 10× NEBuffer r2.1 (New England Biolabs, Ipswich, MA, USA), 2 μL of ssDNA reporter (10 μM), 0.5 μL of recombinant RNase inhibitor (40 U/μL), 2 μL of EnGen Lba Cas12a (Cpf1) (1 μM) from New England Biolabs [44], 1 μL of crRNA, and 2 μL of RT-RPA amplicons, at a total reaction volume of 25 μL. These CRISPR/Cas12a reactions were performed at 37 °C for 15 min. The fluorescence signal was measured at 30 s intervals using a CFX ConnectTM Real-Time PCR System (Bio-Rad Laboratories, Hercules, CA, USA) and confirmed visually under blue and UV light.

### 4.11. Optimization of Reaction Temperature and Time

To determine the optimal temperature and duration for the RT-RPA reactions, we tested temperatures ranging from 34 to 48 °C and confirmed the reaction times by conducting reactions at each temperature for 1, 3, 5, 10, 15, 20, 25, and 30 min.

### 4.12. Evaluation of Specificity and Sensitivity

To assess the specificity of NeLV detection, cross-reactivity was evaluated by analyzing fluorescence intensity and visually observing reactions in samples infected with CymMV, ORSV, TSWV, and CMV, common viral pathogens of *Phalaenopsis* orchids, as well as in healthy samples. To ensure the specificity of the crRNA of the CRISPR/Cas12a system, crRNA-1(5′-UAAUUUCUACUAAGUGUAGAUUCAGGUAAGUUAGGAUGCUUCCC-3′), crRNA-2(5′-UAAUUUCUACUAAGUGUAGAUGGGUAGAACUAACGAUGCCCUUU-3′), and crRNA-3(5′-UAAUUUCUACUAAGUGUAGAUUCAGGCAAAUCCACUUCAACA-3′) was used (Appendix A). Sensitivity was determined by comparing it with RT-PCR using 10-fold dilutions (ranging from 100 fg/μL to 10 zg/μL) of NeLV transcripts. RT-PCR was performed according to the manufacturer’s instructions using SuPrimeScript RT-PCR premix (Genet Bio, Daejeon, Republic of Korea) and forward and reverse primers. Each specificity and sensitivity test was independently repeated three times.

### 4.13. Validation of CRISPR/Cas12 Using Field Samples

To evaluate the performance of RT-RPA-CRISPR/Cas12a in detecting NeLV, 21 *Phalaenopsis* orchid samples and 17 daffodil (*Narcissus tazetta*) samples were used. DEPC-treated water (NTC) and the NeLV transcript were used as negative and positive controls, respectively.

### 4.14. Statistical Analysis

All experiments were repeated three times. Fluorescence intensities for RT-RPA-CRISPR/Cas12a were analyzed using Microsoft Excel 2016 (Microsoft, Redmond, WA, USA) and are expressed as the mean ± SD. The unpaired two-tailed *t*-test was applied for the statistical analysis in Microsoft Excel 2016. Differences with *p* values below 0.05 were considered statistically significant and indicated using the following symbols: * *p* < 0.05; ** *p* < 0.01; *** *p* < 0.001; **** *p* < 0.0001.

## 5. Conclusions

This study highlights the potential of the ONT platform as a valuable tool for identifying and characterizing viral diseases in *Phalaenopsis* orchid samples. Furthermore, our findings illustrate the effectiveness of a CRISPR/Cas12a-based detection assay combined with RT-RPA for the accurate identification of NeLV in both *Phalaenopsis* orchids and daffodils, exhibiting notable levels of specificity and sensitivity. This RT-RPA-CRISPR/Cas12a method can be used to rapidly identify virus-infected samples in certified facilities or to screen virus-free tissue culture plantlets, thereby potentially enhancing the productivity of the orchid industry. Continued efforts in refining sample preparation, optimizing data analysis pipelines, and reducing sequencing costs will be essential to maximize the efficiency and accessibility of nanopore sequencing for plant virology studies. Additionally, the development of multiplexed detection approaches using CRISPR/Cas12a will be critical for expanding its utility in diverse plant species and viral pathogens. Overall, the convergence of nanopore sequencing and CRISPR/Cas12a technology presents exciting opportunities to revolutionize plant virus detection, paving the way for more efficient disease management and crop protection strategies in agriculture. 

## Figures and Tables

**Figure 1 ijms-25-02666-f001:**
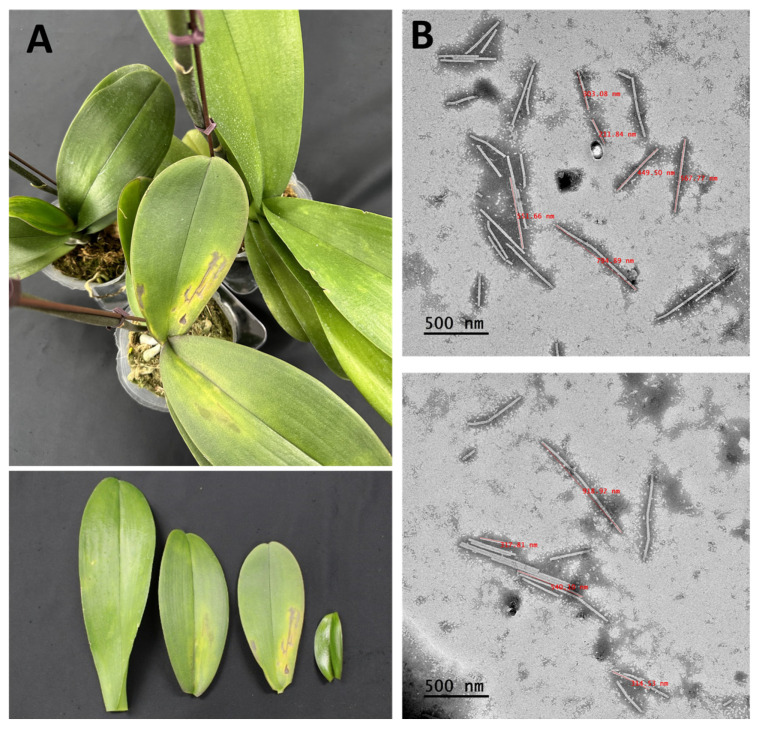
*Phalaenopsis* orchid leaves exhibiting symptoms and the transmission electron microscopy analysis. (**A**) Symptoms of chlorotic flecks, mosaics, and necrosis in *Phalaenopsis* orchids. (**B**) Filamentous virus particles measuring 470–660 nm in length and rod-shaped virus particles of about 300 nm in length observed in the leaf sap of infected *Phalaenopsis* orchids. Scale bars, 500 nm.

**Figure 2 ijms-25-02666-f002:**
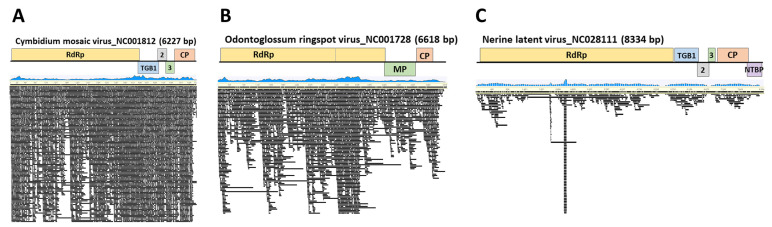
Genome assembly of viral genomes identified through nanopore sequencing. Genomic mapping of (**A**) cymbidium mosaic virus, (**B**) nerine latent virus, and (**C**) odontoglossum ringspot virus found in *Phalaenopsis* orchids. The blue graph represents the mapping distribution of the reads, and the black bars indicate the reads.

**Figure 3 ijms-25-02666-f003:**
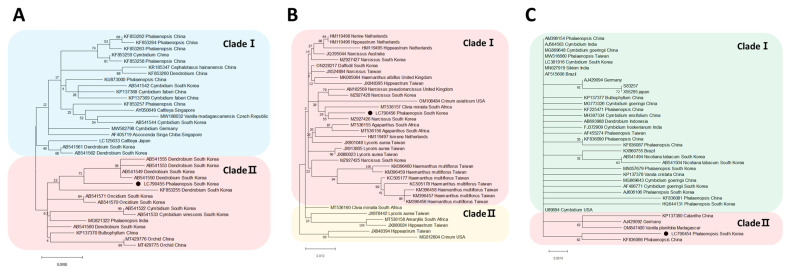
The maximum-likelihood phylogenetic analysis conducted using the coat protein sequences of (**A**) cymbidium mosaic virus, (**B**) nerine latent virus, and (**C**) odontoglossum ringspot virus. The black circles indicate the isolates identified from *Phalaenopsis* orchids.

**Figure 4 ijms-25-02666-f004:**
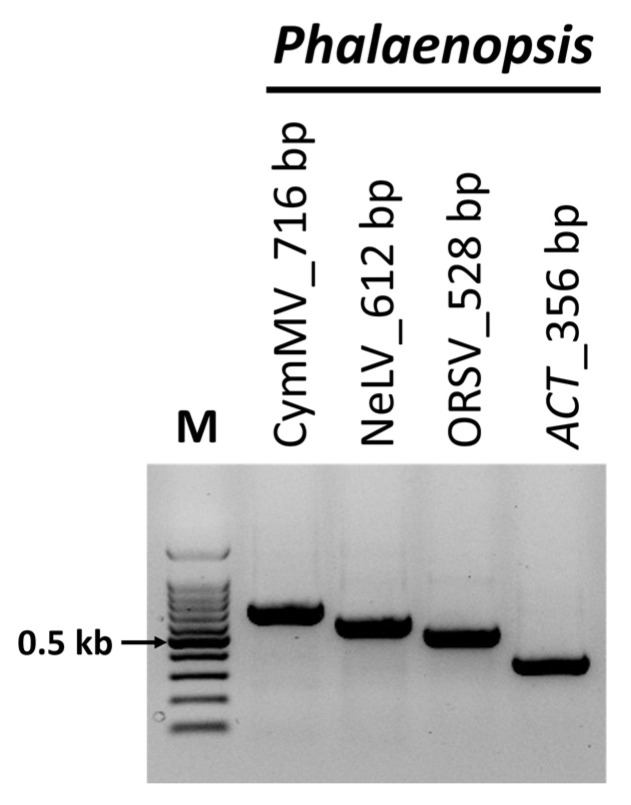
Confirmation of viruses identified through nanopore sequencing using reverse-transcription polymerase chain reaction (RT-PCR). RT-PCR was performed to amplify samples from *Phalaenopsis* orchids using specific primer sets targeting the viruses identified in the nanopore sequencing. *ACT* was used as an internal control. M, 100-bp DNA ladder; ORSV, odontoglossum ringspot virus; NeLV, nerine latent virus; CymMV, cymbidium mosaic virus.

**Figure 5 ijms-25-02666-f005:**
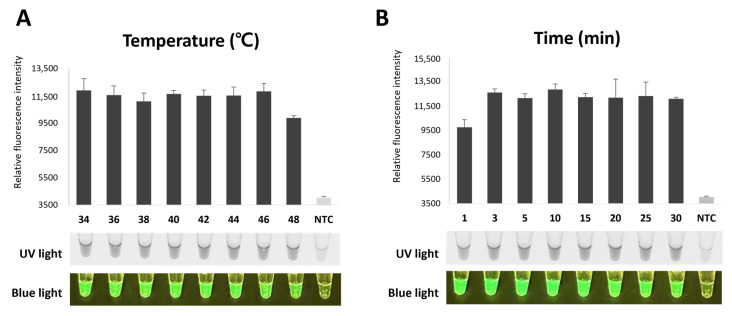
Optimization of RT-RPA-CRISPR/Cas12a for NeLV detection. (**A**) Optimization of reaction temperature for NeLV detection. RT-RPA-CRISPR/Cas12a was performed at a wide range of temperatures, ranging from 34 °C to 48 °C. (**B**) Optimization of reaction time for NeLV detection. RT-RPA-CRISPR/Cas12a amplification can be detected and visualized at 1 min. RT-RPA-CRISPR/Cas12a is visible to the naked eye through fluorescence under UV light and blue light.

**Figure 6 ijms-25-02666-f006:**
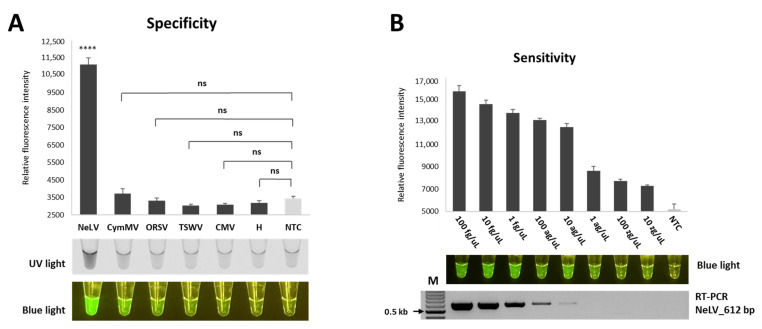
Specificity and sensitivity of RT-RPA-CRISPR/Cas12a in detecting NeLV. (**A**) Evaluation of the specificity of RT-RPA-CRISPR/Cas12a for NeLV detection. NeLV, NeLV-infected template (positive sample); CymMV, CymMV-infected sample; ORSV, ORSV-infected sample; TSWV, TSWV-infected sample; CMV, CMV-infected sample; H, healthy *Phalaenopsis orchids* sample (negative sample); NTC, no template control (water). **** *p* < 0.0001; *p*-value = 0.0000115; ns, not significant. (**B**) Assessments of the sensitivity of RT-RPA-CRISPR/Cas12a for NeLV detection. The detection limits of RT-RPA-CRISPR/Cas12a and RT-PCR were determined using various concentrations of transcripts (100 fg/μL to 10 zg/μL) as templates. The RT-PCR amplification products (612 bp for the RT-PCR) were analyzed using 1.5% agarose gels.

**Figure 7 ijms-25-02666-f007:**
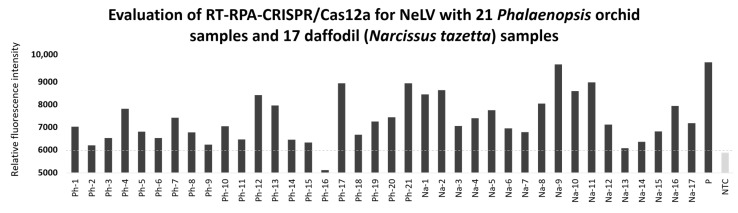
Detection of NeLV in samples of *Phalaenopsis* orchid and daffodil using RT-RPA-CRISPR/Cas12a. Ph-1 to Ph-21, *Phalaenopsis* orchid samples; Na-1 to Na-17, daffodil *(Narcissus tazetta*) samples; P, NeLV-infected template (positive sample); NTC, no template control (water).

**Figure 8 ijms-25-02666-f008:**
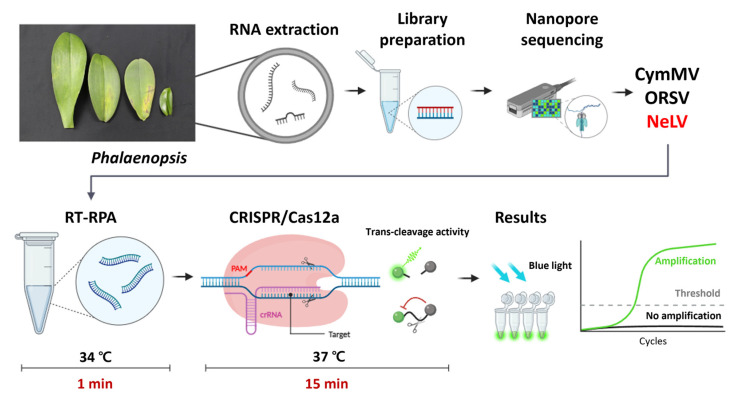
Schematic diagram of nanopore sequencing for the identification of *Phalaenopsis* orchid viruses and the RT-RPA-CRISPR/Cas12a assay used for NeLV detection. Created with Biorender (biorender.com) (accessed on 12 December 2023).

**Table 1 ijms-25-02666-t001:** Data summary from ONT.

Library ID	Raw Reads				Trimmed Reads				SRA Accession Number
	Total Reads	Min Read Length	Max Read Length	Mean Read Length	Total Reads	Min Read Length	Max Read Length	Mean Read Length	
*Phalaenopsis*	528,819	86	17,523	614.2	97,476	1	3466	87.8	SRR22891753

**Table 2 ijms-25-02666-t002:** List of identified viruses in *Phalaenopsis* orchids.

Library ID	Virus Name	Read Number	Maximum Read Length	Mean Read Length	Identity (%)	Coverage (%)	Accession Number
*Phalaenopsis*	Cymbidium mosaic virus	12,913	3466	105.9	92.6	100	LC790455
	Nerine latent virus	1466	2259	62.7	97	94.1	LC790456
	Odontoglossum ringspot virus	2573	2220	134.8	92.3	99.4	LC790454

**Table 3 ijms-25-02666-t003:** Comparison of RT-RPA-CRISPR/Cas12a and RT-PCR for NeLV using *Phalaenopsis* orchid samples and daffodil samples.

**Sample** **name**	**Ph-** **1**	**Ph-** **2**	**Ph-** **3**	**Ph-** **4**	**Ph-** **5**	**Ph-** **6**	**Ph-** **7**	**Ph-** **8**	**Ph-** **9**	**Ph-** **10**	**Ph-** **11**	**Ph-** **12**	**Ph-** **13**	**Ph-** **14**	**Ph-** **15**	**Ph-** **16**	**Ph-** **17**	**Ph-** **18**	**Ph-** **19**	**Ph-** **20**
RT-PCR																			+	+
RT-RPA-CRISPR/Cas12a	+			+			+			+		+	+				+		+	+
**Sample** **name**	**Ph-** **21**	**Na-** **1**	**Na-** **2**	**Na-** **3**	**Na-** **4**	**Na-** **5**	**Na-** **6**	**Na-** **7**	**Na-** **8**	**Na-** **9**	**Na-** **10**	**Na-** **11**	**Na-** **12**	**Na-** **13**	**Na-** **14**	**Na-** **15**	**Na-** **16**	**Na-** **17**	**P**	**NTC**
RT-PCR	+		+			+		+	+	+		+	+	+	+			+	+	
RT-RPA-CRISPR/Cas12a	+	+	+	+	+	+			+	+	+	+	+				+	+	+	

+, represent positive results obtained by each assay; -, represent negative results obtained by each assay.

## Data Availability

The data presented in this study are available in the GenBank database (www.ncbi.nlm.nih.gov) (assessed on 2 January 2024) under the accession numbers listed in the text.

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
