# Peer review of "Identification of Viruses Infecting *Phalaenopsis* Orchids Using Nanopore Sequencing and Development of an RT-RPA-CRISPR/Cas12a for Rapid Visual Detection of Nerine Latent Virus"

_ijms, 2024, doi:10.3390/ijms25052666_

Round 1

Reviewer 1 Report

Comments and Suggestions for Authors

The objective of the paper was found sufficient. The writing of the manuscript is simple and understandable. The language of writing, expression, fluency, and synchronization of the paper is also simple and understandable. I have a few proposals as follows: In the abstract, the topic was addressed directly and the results were given. It is like a conclusion rather than an abstract. However, it seems that it would be better if the full essence of the work was given by talking about the purpose of the study, the importance of the subject and technique, and a few sentences about the method.

Keywords aid in the discoverability of the content of the study. The keywords were the same as the title of the study. However, keywords should be different from the title so that the accessibility of the study can be further increased.

In the last paragraph of the introduction, the results were also given. However, rather than what you did, it would be more accurate to include information about why the study was carried out and what its purpose was.

The conclusion is short. Maybe it should be enriched a little more.

In addition, I have a few minor remarks and suggestions regarding English grammar and writing in lines 3, 34, 42, 49, 50, 184, 260, 318, 356, and 455. I pointed out them in the revised text. 

Comments on the Quality of English Language

 It would be useful to review the English writing of the paper one more time.

Author Response

Reviewer 1

It is my great pleasure to resubmit a revised version of our manuscript (ID: IJMS-2872722) to International Journal of Molecular Science. Many thanks for your decision letter and for allowing us to revise this manuscript. We have carefully considered the reviewer-1 comments and have incorporated all the suggested changes into this revised manuscript. We would like to thank the reviewer-1 for their critical evaluation of this manuscript. Changes in the revised manuscript are indicated by green color. Followings are the major changes included in the revised manuscript:

1. In the abstract, the topic was addressed directly and the results were given. It is like a conclusion rather than an abstract. However, it seems that it would be better if the full essence of the work was given by talking about the purpose of the study, the importance of the subject and technique, and a few sentences about the method.

Response: We revised the abstract and keywords as suggested by reviewer 1. Please see line 12-15.

2. Keywords aid in the discoverability of the content of the study. The keywords were the same as the title of the study. However, keywords should be different from the title so that the accessibility of the study can be further increased.

Response: We revised the keywords as suggested by reviewer 1.

3. In the last paragraph of the introduction, the results were also given. However, rather than what you did, it would be more accurate to include information about why the study was carried out and what its purpose was.

Response: We revised the introduction as suggested by reviewer 1. Please see line 92-95.

4. The conclusion is short. Maybe it should be enriched a little more.

Response: We revised the conclusion as suggested by reviewer 1. Please see line 478-482.

5. In addition, I have a few minor remarks and suggestions regarding English grammar and writing in lines 3, 34, 42, 49, 50, 184, 260, 318, 356, and 455. I pointed out them in the revised text.

Response: As reviewer 1 suggested, we have revised them. Please see the revised manuscript. Thank you.

Reviewer 2 Report

Comments and Suggestions for Authors

The study "Identification of viruses infecting Phalaenopsis orchids using nanopore sequencing and development of a RT-RPA-CRISPR/Cas12a for rapid visual detection of nerine latent virus" presents the utilization of the MinION nanopore sequencing platform for the rapid detection of plant viruses in Phalaenopsis orchids. It reports the identification of three plant viruses and develops a novel RT-RPA-CRISPR/Cas12a assay for the rapid and specific detection of Nerine latent virus (NeLV). The manuscript is well-structured with a comprehensive introduction, clear methodology, and appropriate analysis of results.

The method combining reverse transcription-recombinase polymerase amplification (RT-RPA) with CRISPR/Cas12a for rapid diagnosis appears to be an innovative approach, especially when it is flexible enough to be used with various instruments and offers a high level of accuracy. However, it is not entirely novel as the foundational components, RT-RPA and CRISPR/Cas12a, have been previously used and reported in scientific literature for the detection of different pathogens: The study “Rapid detection of monkeypox virus using a CRISPR-Cas12a mediated assay: a laboratory validation and evaluation study. Open AccessPublished:September 15, 2023DOI:https://doi.org/10.1016/S2666-5247(23)00148-9”

explores a novel diagnostic approach for the rapid detection of monkeypox virus using CRISPR-Cas12a technology. It focuses on developing a sensitive and specific assay capable of identifying monkeypox virus in clinical samples efficiently. This method aims to improve upon traditional diagnostic techniques by offering quicker results and requiring less complex laboratory infrastructure, making it particularly useful in areas with limited access to advanced diagnostic facilities. The method utilizes CRISPR-Cas12a technology for the rapid detection of the monkeypox virus. It involves extracting RNA from samples, converting RNA to DNA (if necessary), amplifying specific viral DNA sequences, and then using a CRISPR-Cas12a based system to recognize and bind to these sequences. Upon recognition, the Cas12a enzyme becomes activated and cuts a reporter molecule, producing a detectable signal. This innovative approach combines the specificity of CRISPR technology with the sensitivity of isothermal amplification, offering a fast, accurate, and potentially field-deployable diagnostic tool.

The RT-RPA technique has been adopted for the detection of various viruses, including plant viruses like Tomato spotted wilt virus, and it's known for its rapid and sensitive detection capabilities that can be conducted without expensive instruments and observable with the naked eye. Such approaches have been discussed for their potential application in resource-limited settings and point-of-care diagnostics, which align with the descriptions of your method​​​​. Moreover, the integration of RT-RPA with lateral flow assays, which seems to be a part of your described method, has been documented before for different viruses and applications, showing its utility in clinical and field diagnostics​​​​.

General Comments:

The manuscript is methodologically sound and provides valuable insights into the application of nanopore sequencing in plant pathology. The development of a CRISPR/Cas12a-based detection assay is not quite innovative and could significantly impact the field. However, there are several areas where the manuscript could be improved to enhance clarity, depth, and overall quality. My major concern is about standard practices to ensure specificity and sensitivity of the assay, i.e. nonsense gRNA Control: This control uses a gRNA that does not target any sequence in the genome of the organism being studied. It helps to demonstrate that any signal or detection observed is due to specific targeting by the functional gRNA and not nonspecific activation of the Cas enzyme.

Mismatch Template Control: Using templates that differ by one or two nucleotides from the target sequence helps to assess the assay's specificity. It shows whether the CRISPR-Cas system can distinguish between highly similar sequences, indicating its ability to accurately identify the target virus even in the presence of closely related sequences.

Including these controls would help validate the assay's performance, ensuring that positive results are due to the presence of the target virus and not due to nonspecific interactions or background noise. Without seeing the specific details of the study, it's challenging to say definitively what might be missing, but incorporating a comprehensive set of controls is crucial for demonstrating the reliability and specificity of any diagnostic method.

Specific Comments and Corrections:

Introduction (Page 2, Lines 52-66):

The introduction could be improved by providing a brief explanation of the importance of rapid and accurate virus detection in orchids and how this impacts commercial cultivation.

others.

Results (Page 7, Lines 577-601):

The results section would benefit from a table summarizing the key findings for quick reference.

The genome coverage statistics are impressive, but the discussion lacks a comparative analysis with other sequencing methods or platforms.

Discussion (Page 8, Lines 1231-1256):

A more detailed discussion on the limitations of the ONT platform and how it might affect the study's outcomes would be helpful (Line 1255).

The potential of the developed RT-RPA-CRISPR/Cas12a assay in broader applications beyond the tested orchids should be discussed (Lines 1231-1256).

Conclusions (Page 12, Lines 892-899):

The conclusions could be more impactful by including future research directions or applications of the study findings.

Figures and Tables:

Ensure that all figures and tables are high quality and legible. For instance, the legends in Figure 8 (Page 9) could be more descriptive.

Statistical Analysis (Page 12, Line 871-877):

Provide specific p-values alongside the symbols for statistical significance.

Consider presenting the data in a figure or table for visual impact and clarity.

Language and Grammar:

The manuscript would benefit from proofreading by a native English speaker, as there are minor grammatical errors throughout the text that could be corrected for better readability.

Author Response

Reviewer 2

It is my great pleasure to resubmit a revised version of our manuscript (ID: IJMS-2872722) to International Journal of Molecular Science. Many thanks for your decision letter and for allowing us to revise this manuscript. We have carefully considered the reviewer-2 comments and have incorporated all the suggested changes into this revised manuscript. We would like to thank the reviewer-2 for their critical evaluation of this manuscript. Changes in the revised manuscript are indicated by red color. Followings are the major changes included in the revised manuscript:

General Comments:

1. My major concern is about standard practices to ensure specificity and sensitivity of the assay, i.e. nonsense gRNA Control: This control uses a gRNA that does not target any sequence in the genome of the organism being studied. It helps to demonstrate that any signal or detection observed is due to specific targeting by the functional gRNA and not nonspecific activation of the Cas enzyme.

Response: As suggested by the reviewer, we confirmed specificity using nonsense gRNA (=crRNA) control, and confirmed amplification only from NeLV-specific crRNA. These results have been added as supplementary Figure 1, line 199-204 and 453-157.

2. Mismatch Template Control: Using templates that differ by one or two nucleotides from the target sequence helps to assess the assay's specificity. It shows whether the CRISPR-Cas system can distinguish between highly similar sequences, indicating its ability to accurately identify the target virus even in the presence of closely related sequences.

I really appreciate the reviewer’s useful comments. However, it’s essential to note that the crRNA for the CRISPR/Cas12a system targeting NeLV was designed from the coat protein region, known for its high conservation. While this region is highly conserved, variation in sequence still exist among different isolates rather than different viruses. Therefore, the CRISPR/Cas12a system should theoretically detect NeLV even with minor differences in the target sequence, such as one or two nucleotide variations. To validate this hypothesis, field sample testing was conducted in this study. To further support these findings, the templates used in field sample testing performed Sanger sequencing, confirming various sequence variants within the crRNA regions. Please see the attached file. Then you can see the supporting image. Really appreciated it. 

Specific Comments and Corrections:

3. Introduction (Page 2, Lines 52-66): The introduction could be improved by providing a brief explanation of the importance of rapid and accurate virus detection in orchids and how this impacts commercial cultivation.

Response: As reviewer 2 suggested, we have revised it. Please see line 57-29.

4. The genome coverage statistics are impressive, but the discussion lacks a comparative analysis with other sequencing methods or platforms.

Response: As reviewer 2 suggested, we have revised it. Please see line 251-256.

5. A more detailed discussion on the limitations of the ONT platform and how it might affect the study's outcomes would be helpful.

Response: As reviewer 2 suggested, we have revised it. Please see line 269-274.

6. The potential of the developed RT-RPA-CRISPR/Cas12a assay in broader applications beyond the tested orchids should be discussed.

Response: As reviewer 2 suggested, we have revised it. Please see line 323-328.

7. The conclusions could be more impactful by including future research directions or applications of the study findings.

Response: As reviewer 2 suggested, we have revised it. Please see line 478-492.

8. Ensure that all figures and tables are high quality and legible. For instance, the legends in Figure 8 (Page 9) could be more descriptive.

Response: As reviewer 2 suggested, we have revised it. Please see Figure 8.

9. Provide specific p-values alongside the symbols for statistical significance.

Response: As reviewer 2 suggested, we have revised it. Please see Figure 6 legend.

10. The manuscript would benefit from proofreading by a native English speaker, as there are minor grammatical errors throughout the text that could be corrected for better readability.

Response: As the reviewer pointed out, the manuscript has been proofread by a native English speaker. Thank you. I have attached a copy of English proofreading certificate. Thank you.

Round 2

Reviewer 2 Report

Comments and Suggestions for Authors

I would like to express my gratitude to the authors for addressing my comments. In my view, additional experiments are not required for publication. However, it would be beneficial to include the three ncrRNAs in "Supplementary Table S1: Information on Primer Sets Used in This Study," with a particular emphasis on their sequences.

Author Response

I would like to express my gratitude to the authors for addressing my comments. In my view, additional experiments are not required for publication. However, it would be beneficial to include the three ncrRNAs in "Supplementary Table S1: Information on Primer Sets Used in This Study," with a particular emphasis on their sequences.
